# Enhanced Optoelectronic Performance and Polarized Sensitivity in WSe_2_ Nanoscrolls Through Quasi-One-Dimensional Structure

**DOI:** 10.3390/nano14231935

**Published:** 2024-11-30

**Authors:** Jinggao Sui, Xiang Lan, Zhikang Ao, Jinhui Cao

**Affiliations:** 1Defense Innovation Institute, Academy of Military Sciences, Beijing 100071, China; js2086@cantab.ac.uk; 2School of Materials Science and Engineering, Hunan University, Changsha 410082, China; lanxiang0901@hnu.edu.cn; 3School of Flexible Electronics (Future Technologies), Institute of Advanced Materials, Jiangsu National Synergetic Innovation Center for Advanced Materials, Nanjing Tech University, Nanjing 211816, China; iamzkao@njtech.edu.cn; 4College of Energy and Power Engineering, Changsha University of Science and Technology, Changsha 410114, China

**Keywords:** transition metal dichalcogenides, quasi-one-dimensional structure, nanoscroll, polarized sensitivity

## Abstract

Transition metal dichalcogenides (TMDs), such as tungsten diselenide (WSe_2_), are expected to be used in next-generation optoelectronic devices due to their unique properties. In this study, we developed a simple method of using ethanol to scroll monolayer WSe_2_ nanosheets into nanoscrolls. These nanoscrolls have a quasi-one-dimensional structure, which enhances their electronic and optical properties. The characterization confirmed their unique structure, and the photodetectors made of these nanoscrolls have high sensitivity to polarized light, with anisotropy ratios of 1.3 and 1.7 at wavelengths of 638 nm and 808 nm. The enhanced light response is attributed to the large surface area and quantum wire-like behavior of the nanoscrolls, making them suitable for advanced polarization-sensitive devices.

## 1. Introduction

Two-dimensional (2D) materials, particularly transition metal dichalcogenides (TMDs), have garnered significant attention in recent years due to their tunable electronic, optical, and mechanical properties, which are crucial for various applications in nanoelectronics and optoelectronics. Among these, tungsten diselenide (WSe_2_) stands out due to its high carrier mobility [1], strong spin–orbit coupling [2], and direct bandgap in the monolayer form, making it an excellent candidate for photodetectors, transistors, and other electronic devices.

Recent advancements in the synthesis of 2D materials have enabled the exploration of new structural configurations, such as nanoscrolls [3,4,5], which exhibit quasi-one-dimensional characteristics. In particular, the formation of WSe_2_ nanoscrolls introduces significant quantum confinement effects, akin to those observed in quantum wires. When the dimensions of a material are reduced to the nanoscale, specifically in two dimensions, charge carriers experience confinement, leading to discrete energy levels and altered electronic properties [6]. This quantum wire-like behavior in WSe_2_ nanoscrolls is expected to enhance light–matter interaction, improve charge carrier mobility, and increase the material’s overall optoelectronic performance.

In this study, we report a simple and effective method for fabricating WSe_2_ nanoscrolls from monolayer nanosheets via an ethanol-induced rolling process. This approach leverages the residual stresses introduced during the growth process and the thermal expansion mismatch between the nanosheet and the substrate. The resulting WSe_2_ nanoscrolls were thoroughly characterized using optical microscopy, transmission electron microscopy (TEM), Raman spectroscopy, and photoluminescence (PL) spectroscopy to understand their morphology and structural properties. Additionally, we explore the optoelectronic performance of photodetectors based on these nanoscrolls, with a particular focus on their polarized sensitivity under different wavelengths of light. The quantum confinement effects, when their dimensions are reduced below 100 nm, observed in these nanoscrolls are expected to play a significant role in their enhanced photoresponse, making them promising candidates for next-generation polarization-sensitive photodetectors.

## 2. Materials and Methods

### 2.1. Growth of Monolayer WSe_2_

A quartz boat containing WSe_2_ powder (~5 g) is located in the central heating zone of the furnace as the growth source, and clean SiO_2_ (~285 nm)/Si is used as the growth substrate in the furnace. The horizontal distance between the powder and the substrate should be maintained between 15 and 18 cm. During the process of temperature rise, the substrate is upstream of the source, and the Ar_2_ airflow flows from the substrate towards the source. The airflow is set to 250 sccm and maintained for 15 min to remove undesired oxygen and water vapor. Then, we reduce the Ar_2_ gas flow to 75 sccm. When the target temperature of 1160 °C is reached in the central heating zone, the flow direction of the airflow is reversed, and the airflow flows from the source to the substrate. The substrate changes from being located upstream of the source to being located downstream of the source, and then remains for 5 min for growth. Next, we reverse the direction of the airflow again and stop heating to halt the growth process and allow it to cool naturally to room temperature. The entire process is carried out under ambient pressure in an argon atmosphere.

### 2.2. Fabrication of WSe_2_ Nanoscroll

A drop of ethanol is added onto a substrate grown with monolayer WSe_2_ nanosheets to fabricate WSe_2_ nanoscrolls. It should be noted to ensure that the nanosheets are completely covered by the ethanol solution. When ethanol infiltrates the interface between the nanosheets and the substrate, a portion of nanosheet detaches from the substrate. Subsequently, the detached part of the nanosheet scrolls in the solution and drives the remaining parts, ultimately forming a nanoscroll. The obtained nanoscrolls are further dried under ambient temperature Ar_2_ airflow to remove residual solutions on the substrate and the nanoscrolls.

### 2.3. Material Characterization

Optical microscopes and TEM are used to characterize the morphology of nanocarriers. Raman spectroscopy is performed at room temperature using a Renishaw LabRAM Invia micro-Raman system (Renishaw plc, Shanghai, China) excited by a 488 nm laser. The applied laser power is set to 5%, and the exposure time is approximately 0.1 s. Transmission electron microscopy (TEM) is taken with a JEOL JEM−F200 (JEOL Ltd., Tokyo, Japan).

### 2.4. Device Fabrication and Measurement

WSe_2_ nanoscroll devices are manufactured on a SiO_2_/Si substrate. Firstly, 50 nm thick Au electrodes are deposited on the SiO_2_/Si substrate by using standard photolithography and high-vacuum electron beam evaporation processes. Then, we coat the substrate with a layer of hexamethyldisilazane (HMDS) to enhance adhesion, and spin coat with polymethyl methacrylate (PMMA). Due to the pre-functionalization of HMDS, the adhesion between the PMMA layer and the SiO_2_/Si substrate is weak, and mechanical release is achieved by wrapping the metal electrode with a heat release tape. Then, we precisely align the released metal electrodes under an optical microscope and laminate them onto the WSe_2_ nanoscroll to ensure clean contact between the metal and the sample. Finally, the PMMA covering the contact pads is removed using traditional electron beam lithography, and the exposed metal pads are used for performance measurement.

The photoinduced current of the WSe_2_ nanoscroll devices was measured using a highly sensitive source meter capable of detecting currents in the nanoampere range. The sample was illuminated by lasers with wavelengths of 638 nm and 808 nm at controlled optical power densities, and the photocurrent was recorded under a constant bias voltage of 1 V. The source meter was connected in series with the device, and the incident light was modulated to precisely measure the photocurrent response. The experimental environment was carefully controlled to eliminate external noise, ensuring accurate detection of nanoampere-scale photocurrents. The optical power density was kept at 72.8 mW/cm^2^, and the photocurrent was measured across a range of polarization angles to assess anisotropy.

## 3. Results

### 3.1. Rolling up WSe_2_ Nanosheet into Nanoscroll

In typical experiments, large-scale monolayer TMD nanosheets were synthesized on SiO_2_/Si substrates using a reverse-flow CVD technique [7]. Then, a droplet of ethanol solution (volume ratio of ethanol/water = 3:1) was applied onto WSe_2_ nanosheets to rapidly and efficiently fabricate WSe_2_ nanoscrolls. Figure 1 provides the preparation process of WSe_2_ nanoscrolls. Briefly, when ethanol solution is embedded at the interface between nanosheets and substrate, due to the mismatch in thermal expansion coefficient between nanosheets and substrate, a portion of the nanosheets will be released. The monolayer nanosheets grown by vapor deposition introduce residual stress during the cooling process to room temperature, which causes the nanosheets to spontaneously roll up into nanoscrolls [8]. In our experiment, we found that by tilting the substrate and stopping the rolling process in the middle state, the ethanol solution can be quickly removed, while the layers are driven to roll up. But, the rolling process of nanoscrolls usually occurs within 1 s, making it difficult to control. The number of winding layers in the nanoscroll is determined by the time elapsed between the application of ethanol and the tilted substrate. It is worth noting that we have noticed that due to the limited rolling diameter and size of the nanosheets, the nanoscroll can be restricted within 100 nm on the *z*-axis and *y*-axis (the direction along the long axis of the nanoscroll is usually defined as the *x*-axis), allowing charge carriers to move freely in the *x*-axis direction, similar to the movement of charge carriers in carbon nanotubes or silicon nanowires [9,10]. This indicates that the nanoscrolls we prepared can exhibit quasi-one-dimensional properties similar to quantum wires under certain conditions [11].

### 3.2. Characterization of Monolayer WSe_2_ Nanoscroll

To observe the morphology, surface, and microstructure of the obtained samples, optical microscopy and TEM studies were conducted. Figure 2a shows the optical image of WSe_2_ nanoscrolls transformed from monolayer WSe_2_ nanosheets. To better demonstrate the differences between nanosheets and nanoscrolls, we chose to place a 1D/2D homogeneous junction where nanosheets and nanoscrolls coexist, represented as a schematic diagram. Due to the thickness difference between WSe_2_ nanosheets and nanoscrolls, the nanoscrolls exhibit a thicker middle region and thinner ends, resulting in a slightly different optical contrast [12]. The white box displays the initial size of the nanoscroll, while the black arrow clearly indicates the rolling direction of the nanoscroll. Here, we can control the rolling process to ensure that the height of the nanosheets is below 100 nm after conversion into nanoscrolls, which means that nanoscale quantum wires are formed here [13]. To further explore the spatial modulation of structural and optical properties in the rolling structure of nanosheets, Raman confocal microscopy was used to characterize the nanoscroll structure. We selected two points labeled as “1” and “2” in Figure 2a for the Raman study, where the “1” (red point) and “2” (blue point) are located in the nanosheet and nanoscroll regions, respectively. The Raman spectra at different positions show significant differences: the Raman spectra of the nanosheet region have two significant peaks at 247.2 cm^−1^ and 258.7 cm^−1^ (red line in Figure 2b), corresponding to the E^1^_2g_ and A_1g_ resonance modes of WSe_2_. There is a significant peak in the nanoscroll region at 247.2 cm^−1^ (blue line in Figure 2b), which is consistent with the vibration mode of WSe_2_ at E^1^_2g_, but the Raman signal at A_1g_ disappears. This is because the nanoscroll has a structure similar to multilayer nanosheets [14]. The PL spectral characterization confirmed this. The nanosheet region has a significant PL (at 1.65 eV), but the PL intensity in the nanoscroll region decreases significantly, and the PL peak shows a red shift of 20 meV. The PL peak is an indicator of changes in electronic structure, which can be explained by the stacking and tension effects of the nanoscroll [15,16]. At the same time, we found that the Full Width at Half Maximum (FWHM) PL peak of the nanoscroll broadened significantly. The calculated FWHM of the nanosheet is 0.072, while the nanoscroll shows a broader FWHM of 0.101, which may attribute to the signal overlap of multilayer WSe_2_ and strain effect [17]. This physical and electronic structural change indicates that the nanoscrolls have significant potential for application in the field of micro low-power devices.

Subsequently, TEM studies were conducted on the prepared nanoscrolls (Figure 3). The low-magnification TEM image shows that the nanoscroll is uniformly rolled up (Figure 3a). Figure 3b shows a high-resolution transmission electron microscopy (HRTEM) image of the nanoscroll winding structure (marked by the red box in Figure 3a), while revealing the layered stacking structure of the nanoscroll to be relatively tight. The distance between adjacent layers is about 0.65 nm, which is consistent with the distance between layers of the bulk TMD crystal [18]. The selected area diffraction pattern (SAED) displays the hexagonal lattice symmetry of WSe_2_ nanoscrolls, confirming the preservation of the ordered hexagonal crystal structure (Figure 3c). Multiple diffraction patterns are attributed to the lattice constant changes caused by strain in different layers of multilayer nanosheets, resulting in complex diffraction patterns. The red and yellow hexagonal boxes correspond to lattice constants of 0.34 nm and 0.35 nm (the pristine lattice constant of WSe_2_ is 0.33 nm), respectively. The distance between the spots further confirms the strain difference caused by the rolling of the nanoscroll, with a significant lattice strain.

### 3.3. Performance Characterization of Monolayer WSe_2_ Nanoscroll

Photoresponse is an important indicator for evaluating the performance of photodetectors [19,20]. Figure 4a shows a schematic diagram of a photodetector based on WSe_2_ nanoscrolls. Figure 4b,c show the photoresponse curves of a photodetector based on WSe_2_ nanoscrolls measured at wavelengths of 638 nm and 808 nm, respectively. The detection value of the photoelectric detector based on WSe_2_ nanoscrolls under red laser is 1.5 nA. Interestingly, the corresponding photocurrent is as high as 1.3 nA at 808 nm, indicating that our prepared nanoscroll also has excellent photoresponsiveness in the near-infrared band. The excellent light response of WSe_2_ nanoscrolls can be explained by the curled structure having a large specific surface area. This unique geometric shape enhances the interaction between light and materials, allowing more light to be captured and to excite electron–hole pairs, thereby improving light response efficiency. When the nanoscrolls enter the nanoscale in two dimensions, the movement of charge carriers is restricted, which may lead to higher light absorption and stronger photocurrent response, thereby further enhancing the photoresponsive properties [21].

To explore the polarized sensitivity of photodetectors based on monolayer WSe_2_ nanoscrolls, angle-resolved photocurrent was systematically studied [22,23,24]. Figure 5a shows a schematic diagram of the polarization optoelectronic device of WSe_2_ nanoscrolls. The output current was measured by adjusting the polarization direction of the incident light (Figure 5b,c). Under a constant bias voltage of 1 V, the direction along the long axis of the nanoscroll is defined as 0°, ignoring the directionality of the current. It is observed that the maximum photocurrent occurs when the incident polarized light is almost parallel to the nanoscroll (0°and 180°). When the incident polarized light is perpendicular to the nanoscroll (90°and 270°), the photocurrent reaches its minimum value. This observation corresponds to the anisotropic linear dichroism of the material. The periodic pattern of the nanoscroll was displayed by fitting the data with a sine function to reveal the trend of current variation. A clear comparison indicates that the output current of the nanoscroll photodetector varies periodically with the polarization direction of the incident light. The photocurrent anisotropy ratios of the nanocarrier photodetector under 638 nm and 808 nm irradiation are 1.2 and 1.5, respectively. These results indicate that photodetectors based on WSe_2_ nanoscrolls exhibit a highly polarization-sensitive light response.

## 4. Conclusions

We successfully rolled monolayer WSe_2_ nanosheets into nanoscrolls using an ethanol-assisted technique. These nanoscrolls, with their quasi-one-dimensional structure, enhanced their optoelectronic performance and showed quantum confinement effects with scaling in two dimensions below 100 nm. Photodetectors based on these nanoscrolls demonstrated high sensitivity to polarized light, with notable anisotropy ratios at different wavelengths. These findings highlight the potential of WSe_2_ nanoscrolls for advanced optoelectronic devices, particularly those requiring high polarization sensitivity and benefiting from quantum confinement in nanoscale materials.

## Figures and Tables

**Figure 1 nanomaterials-14-01935-f001:**
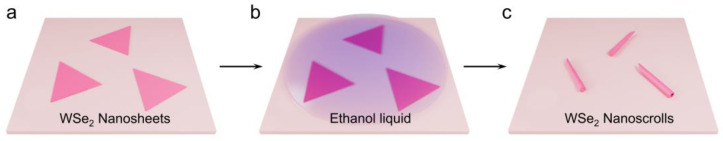
Fabrication of monolayer WSe_2_ nanoscroll process. (**a**) The WSe_2_ nanosheets are grown on the substrate. (**b**) A droplet of ethanol solution is placed on the substrate and cover with the nanosheets, (**c**) The WSe_2_ nanosheets scroll into the nanoscrolls.

**Figure 2 nanomaterials-14-01935-f002:**
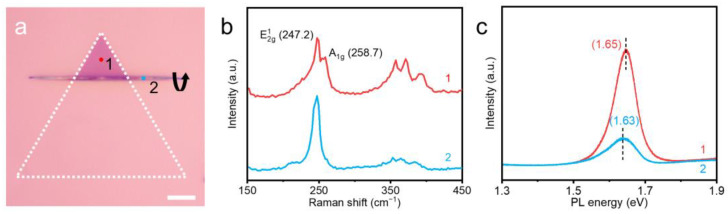
(**a**) Optical microscope image of monolayer nanoscroll. The white dashed triangle box represents the pristine nanosheet. The black arrow represents the rolling direction of the nanoscroll. The red and blue points correspond to the flake the nanoscroll regions, respectively. (**b**,**c**) Raman and PL spectra corresponding to the red and blue points marked in (**a**). Scale bar, 5 µm.

**Figure 3 nanomaterials-14-01935-f003:**
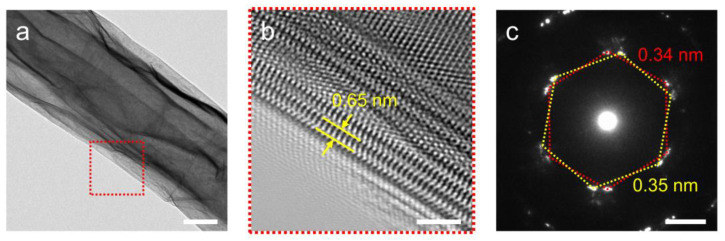
(**a**) Low-magnification TEM image of the nanoscroll. Scale bar, 200 nm. (**b**) Magnified HRTEM image of the nanoscroll selected from the red dashed box in (**a**). Scale bar, 2 nm. (**c**) The SAED pattern of the WSe_2_ nanoscroll. The red and yellow dashed boxes correspond to patterns of the different layers. Scale bar, 1/2 nm.

**Figure 4 nanomaterials-14-01935-f004:**
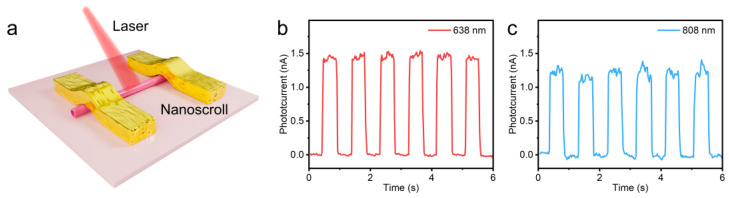
Optoelectronic performance of the monolayer WSe_2_ nanoscrolls. (**a**) Schematic diagram of monolayer WSe_2_ nanoscroll photodetector. (**b**,**c**) The photoresponse curves of the WSe_2_ nanoscrolls under the wavelengths of 638 nm and 808 nm, respectively.

**Figure 5 nanomaterials-14-01935-f005:**
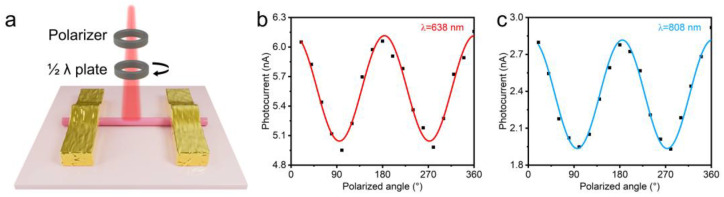
Polarization-sensitive optoelectronic performance of monolayer WSe_2_ nanoscrolls. (**a**) Schematic diagram of polarized optoelectronic devices based on monolayer WSe_2_ nanoscrolls. (**b**,**c**) Plots of the relationship curves between the photocurrent and the polarized angle under the wavelengths of 638 nm and 808 nm, respectively.

## Data Availability

The data generated within this study and the samples related to this study are available from the corresponding author upon reasonable request.

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
