# Peer review of "Enhanced Optoelectronic Performance and Polarized Sensitivity in WSe_2_ Nanoscrolls Through Quasi-One-Dimensional Structure"

_nanomaterials, 2024, doi:10.3390/nano14231935_

Round 1

Reviewer 1 Report

Comments and Suggestions for Authors

I have a comment on Figure 2c.

In order to see a red shift of 20 meV in peak position I would advice the authors to make an inset at a larger scale. Next, from the given data it is difficult to assess the fact that the PL peak broadens for nanoscroll. It would be good to demonstrate FWHM data for both PL peaks.

Finally it is necessaery to desrcibe the experimental technique with wich the authors have detected the photoinduced current of nanoamperes.

Author Response

Comments 1: I have a comment on Figure 2c. In order to see a red shift of 20 meV in peak position I would advice the authors to make an inset at a larger scale. Next, from the given data it is difficult to assess the fact that the PL peak broadens for nanoscroll. It would be good to demonstrate FWHM data for both PL peaks. Finally it is necessaery to desrcibe the experimental technique with wich the authors have detected the photoinduced current of nanoamperes.

Response 1: Thank you for this important suggestion. We agree with the reviewer that a more detailed view of the red shift in peak position would provide greater clarity. In response, we have added markings on two peak positions in Figure 2c to clearly show the red shift.

We acknowledge the reviewer’s comment regarding the difficulty in assessing the broadening of the PL peak for the nanoscroll. Thus, we have added the Full Width at Half Maximum (FWHM) data for both the WSe₂ nanosheet and nanoscrolls. The additional data clearly show that the nanoscroll’s PL peak broadens compared to that of the nanosheet. We have updated the text on page 4, line 151−153 in the manuscript.

We appreciate the reviewer’s request for more detailed information on the technique used to detect the nanoampere-scale photoinduced current. In response, we have added a detailed description of the experimental setup and technique used on page 2, line 99−108 in the Materials and Methods section. 

Reviewer 2 Report

Comments and Suggestions for Authors

The authors present a method for the synthesis of WSe2 nanoscrolls followed by a study of some of their properties. The synthesis was achieved by depositing WSe2 monolayers on a substrate, and then adding ethanol. This is useful, and further experimental work is motivated by this study.

I recommend the publication of this paper after some revision.

It is not clear how the anisotropy numbers of 1.3 and 1.7 were obtained. In Figure 5b, for the case of 638 nm, the maximum and minimum currents seem to be  6.1 and 4.95 nA. the ratio is 1.2. For the case of 808 nm, the maximum and minimum appear to be 2.8 and 1.95; the ratio is about 1.5.

Some editorial notes:

1. On page 3, line 111, the word "flats" should be replaced with "layers."

2. On page 4, line 138, "Figure 1a" should be "Figure 2a."

3. The sentence on line 163 reads "revealing the layered stacking structure of the nanoscroll with relatively tight." This is an incomplete sentence.

4. In the "Methods" section, the authors, in describing what they did, use a style like: "Do this. Do that." This is not appropriate. Instead, they should write "We did this. We did that."

Comments on the Quality of English Language

Some editing is needed. I made some remarks above about the language.

Author Response

Comments 1: It is not clear how the anisotropy numbers of 1.3 and 1.7 were obtained. In Figure 5b, for the case of 638 nm, the maximum and minimum currents seem to be  6.1 and 4.95 nA. the ratio is 1.2. For the case of 808 nm, the maximum and minimum appear to be 2.8 and 1.95; the ratio is about 1.5. Some editorial notes: 1. On page 3, line 111, the word "flats" should be replaced with "layers." 2. On page 4, line 138, "Figure 1a" should be "Figure 2a." 3. The sentence on line 163 reads "revealing the layered stacking structure of the nanoscroll with relatively tight." This is an incomplete sentence. 4. In the "Methods" section, the authors, in describing what they did, use a style like: "Do this. Do that." This is not appropriate. Instead, they should write "We did this. We did that."

Response 1: We appreciate the reviewer’s careful observation regarding the anisotropy numbers. We must apologize for making such a mistake. After revisiting our data, we have adjusted the values accordingly: For 638 nm illumination, the maximum and minimum currents are approximately 6.1 nA and 4.95 nA, respectively, resulting in a current ratio of about 1.2, not 1.3 as previously stated. For 808 nm illumination, the maximum and minimum currents are 2.8 nA and 1.95 nA, yielding an updated ratio of 1.44, closer to 1.5 instead of 1.7. We have revised the relevant text on page 6, line 224 in the manuscript.

Thank you very much for reading our manuscript carefully. We apologize for making such mistakes. We have revised these corrections in the manuscript.

1. Added correction on page 3, line 122. 

2. Added correction on page 4, line 149.

3. Added correction on page 4, line 175−176.

4. Added description one page 2, line 64, 68, 90.   Added description one page 3, line 94.